# Low level SARS-CoV-2 RNA detected in plasma samples from a cohort of Nigerians: Implications for blood transfusion

**Azuka Patrick Okwuraiwe**[1]*, **Chika Kingsley Onwuamah**[1], **Joseph Ojonugwa Shaibu**[1], **Samuel Olufemi Amoo**[1], **Fehintola Anthonia Ige**[1], **Ayorinde Babatunde James**[2], **Leona Chika Okoli**[1], **Abul-Rahman Ahmed**[1,3], **Jamda Ponmak**[1,4], **Judith O. Sokei**[1], **Sulaimon Akanmu**[5], **Babatunde Lawal Salako**[6], **Rosemary Ajuma Audu**[1]

**1** Centre for Human Virology and Genomics, Microbiology Department, Nigerian Institute of Medical Research, Yaba, Lagos, Nigeria, **2** Biochemistry Department, Nigerian Institute of Medical Research, Yaba, Lagos, Nigeria, **3** Department of Cell Biology and Genetics, University of Lagos, Akoka, Lagos, Nigeria, **4** Federal College of Veterinary and Medical Laboratory Technology, Vom, Plateau State, Nigeria, **5** Department of Haematology, College of Medicine, University of Lagos, Idi-Araba, Lagos, Nigeria, **6** Clinical Sciences Division, Nigerian Institute of Medical Research, Yaba, Lagos, Nigeria

* azukaokwu@yahoo.com, apokwuraiwe@nimr.gov.ng

**Data Availability Statement:** All relevant data are within the paper and its Supporting information files.

## Abstract

The present global pandemic triggered by the severe acute respiratory syndrome coronavirus 2 (SARS-CoV-2), has lingered for over a year in its devastating effects. Diagnosis of coronavirus disease 2019 (COVID-19) is currently established with a polymerase chain reaction (PCR) test by means of oropharyngeal-, nasopharyngeal-, anal-swabs, sputum and blood plasma. However, oral and nasal swabs are more commonly used. This study, therefore, assessed sensitivity and specificity of plasma as a diagnostic in comparison with a combination of oral and nasal swab samples, and the implications for blood transfusion. Oropharyngeal (OP) and nasopharyngeal (NP) swab samples were obtained from 125 individuals suspected to have COVID-19 and stored in viral transport medium (VTM) tubes. Ten millilitres of blood samples in EDTA were also obtained by venepuncture and spun to obtain plasma. Viral RNA was obtained from both swabs and plasma by manual extraction with Qiagen QIAamp viral RNA Mini Kit. Detection was done using a real time fluorescent RT-qPCR BGI kit, on a QuantStudio 3 real-time PCR instrument. Average age of study participants was 41 years, with 74 (59.2%) being male. Out of the 125 individuals tested for COVID-19, 75 (60%) were positive by OP/NP swab. However, only 6 (4.8%) had a positive plasma result for COVID-19 with median Ct value of 32.4. Sensitivity and specificity of RT-PCR SARS-CoV-2 test using plasma was 8% and 100% respectively. There was no false positive recorded, but 69 (55.2%) false negatives were obtained by plasma. SARS-CoV-2 viral RNA was detected, albeit low (4.8%) in plasma. Plasma is likely not a suitable biological sample to diagnose acute SARS-CoV-2 infection. The implication of transfusing blood in this era of COVID-19 needs further investigations.

**Funding:** The authors received no specific funding for this work.

**Competing interests:** The authors have declared that no competing interests exist.

# Introduction

Coronavirus disease 2019 (COVID-19) is an existing public health crisis menacing the globe and affecting at least 213 countries for over a year [1]. The virus likely originated in bats and was spread to humans through yet unidentified intermediary animals in Wuhan, Hubei province, China in December 2019 [1]. Globally as of 18[th] February, 2021, there has been 110,520,533 cases, 85,417,829 recoveries and 2,442,945 deaths. The first confirmed case in Nigeria was reported on 27[th] February 2020 in Lagos State and has since been reported in almost all states of the country. There have been 149,369 cases, 1,787 deaths and 125,722 recoveries as of 18[th] February 2021.

Severe acute respiratory syndrome-coronavirus-2 (SARS-CoV-2), is highly infectious, and fears have stood high around the most suitable sample type and testing platform available and their efficacy in containing the pandemic [2]. Currently, the gold standard of diagnosis of the disease is by PCR using the oropharyngeal (OP) and/or nasopharyngeal (NP) swabs. Samples for COVID-19 have also been collected from sputum, tracheal aspirate, saliva, bronchoalveolar lavage, stool, urine and blood. NP swab and/or OP swab are often endorsed for screening or diagnosis of early infection, and is practised in most collection centres in Nigeria [3–5]. A single NP swab is the preferred swab, as it is endured better by the patient and safer for the health worker collecting the sample. NP swabs have an intrinsic quality control. When used properly it reaches the back end of the nasal cavity [5].

After collection, swabs are placed in a universal viral transport medium (VTM) for swift conveyance to the laboratory, ideally under refrigerated conditions [6]. It should be noted, that in some cases, NP or OP swabs may miss early infection, and, repeated testing or obtaining lower respiratory tract specimens may be required.

However, there have been some reports that a negative result [7, 8] does not necessarily exclude infection. Other samples that have been utilised for molecular diagnosis include anal swab and blood plasma. The likelihood of detecting SARS-CoV-2 from these samples vary from one individual to another and it may change during the course of a patient's illness. Negative nasal or oropharyngeal samples have been observed in patients with pneumonia but positive sputum samples [9].

A real-time reverse transcriptase-polymerase chain reaction (RT-PCR) is the World Health Organization (WHO) recommended method for molecular testing for COVID-19 [10–12]. Molecular detection technique needs a degree of proficiency and laboratory requirements, which may limit extensive use with consequent under testing for of COVID-19 diagnosis. RT-PCR assays have the major benefit of amplification and analysis being done concurrently in a closed system. This minimizes false-positive results associated with amplification product contamination.

Although coronaviruses usually infect the upper or lower respiratory tract, viral shedding in plasma or serum is common. There is a potential risk of COVID-19 transmission through transfusion of blood products [13]. There is a large proportion of asymptomatic infections in COVID-19 cases; therefore, considerations of blood safety have arisen especially in endemic areas.

Nigeria (like most countries) has adopted OP/NP swabs for sample collection, so it would be necessary to relate outcomes obtained with corresponding plasma. Testing blood samples for SARS-CoV-2 is also important as it could affect health policy on blood transfusion services, more especially with community transmission and asymptomatic infection prevailing. It is imperative to identify infected individuals, isolate and treat them to prevent transmission of the virus. Inability to identify cases due to sample unsuitability could weaken efforts to contain the current outbreak.

## Aims of the study

The study seeks to compare molecular real-time qPCR SARS-CoV-2 testing from paired oral/nasal swabs and plasma; and assess the sensitivity and specificity of plasma as a sample type for COVID-19. Findings would aid further understanding of the viral pathogenesis, and inform labile blood products use.

## Methods

The Nigerian Institute of Medical Research IRB approved the study with IRB number IRB/20/020.

The form of consent obtained was written form.

### Study design

This was a cross-sectional study conducted within three months, from April to June 2020.

### Study population and location

The Nigerian Institute of Medical Research (NIMR), Yaba, Lagos, with other collaborating organisations, in response to the COVID-19 outbreak, designed a sample collection centre (modified drive-through) in its premises for suspected cases of COVID-19. People who feel uncertain that they may have the infection, filled an online application on the institutional website, and were invited for sample collection. Study participants were drawn from amongst those who visited the centre for testing. The online form had sections inquiring about presence of symptoms. The criteria for inviting individuals for testing after online registration included presenting with fever, sore throat, cough or breathing difficulty, travel history from COVID-19 hotspots, amongst other symptoms/indices.

### Ethical consideration: Consent documentation

Prior to study initiation, ethical approval was obtained from the Institutional Review Board of NIMR, and only those who consented online were enrolled for the study, to cover collecting samples for diagnostic and research purposes.

### Sample collection

Samples were taken from consenting symptomatic and majorly asymptomatic persons. Those who declined blood collection were excluded from the study. Complete personal protective equipment (PPE) was donned before collecting samples from each suspected case. OP and NP swabs along with blood samples were collected. The swab samples were collected by inserting swabs to the end of the nasal and throat regions respectively, swirling the area for 10 seconds and then immediately immersing the swabs into the same tube containing 2 ml of VTM. In addition, venous blood samples were collected in EDTA-anticoagulated tubes. All samples were transported in cold chain (2–8˚C) to the Centre for Human Virology and Genomics (CHVG) of NIMR for analysis.

### Laboratory methods

The laboratory testing was performed at NIMR's ISO 15189:2012 accredited CHVG, between April to June 2020. The Centre is also a WHO prequalification evaluating laboratory.

The blood samples (about 5 ml) in EDTA-anticoagulated tubes were centrifuged at 4000 rpm for 20 minutes to obtain plasma. Viral RNA was extracted from 200µl of oral and nasal

swabs as well as from 200μl plasma using the QIAamp viral RNA Mini Kit (Qiagen, Hilden, Germany). Viral RNA was subsequently amplified and detected utilising a real-time fluorescent RT-PCR kit (Beijing Genomics Institute (BGI), Shenzhen, China) to detect SARS-CoV-2, according to manufacturers' instructions.

**RT-qPCR.** One-step reverse transcriptase (RT) real-time (qPCR) was carried out to detect SARS-CoV-2 using qPCR assay. The process contained 18.5μl of nucleic acid mix, 1.5μl of enzyme mix and 10μl of RNA in a total reaction volume of 30μl. RT-qPCR cycling was performed on QuantStudio 3 (Applied Biosystems) as follows: 50°C for 20 minutes, 95°C for 10 minutes, then 40 cycles of 15 seconds at 95°C and 30 seconds at 60°C.

The BGI assay detects the SARS-CoV-2 ORF1 region, with an internal control detecting a human housekeeping gene [14]. The internal control is to verify that sampling was correctly taken and checks for false negative or inhibition. Assay validation included ensuring curves are S-shaped, no cycle threshold (Ct) values for the negative control, and both targets detected for the positive control with Ct $\leq$ 32 [14]. All samples tested had the internal control detected at Ct $\leq$ 32 to be accepted as valid, and the assay has a limit of detection of 100 copies/ml [14]. Only results from the oral and nasal swabs were issued out to requesting individuals, according to national testing guidelines for COVID-19.

**Data analysis.** Data was entered into Microsoft Excel software (Microsoft Corporation, Redmond Washington, USA), and statistical analysis was done on SPSS v22 (IBM, Chicago, IL, USA).

## Results

A total of 125 prospectively collected paired NP and OP swabs were used. These samples also had accompanying blood samples taken over a period of 3 months (April to June, 2020). There were 74 (59.2%) male and 51 (40.8%) female individuals. The mean age of the individuals was 41 years, with a range of 18 to 72 years. A total of 75 (60%) persons had laboratory-confirmed SARS-CoV-2 RNA infection from NP/OP swabs. From the corresponding plasma samples, 6 (4.8%) were positive for SARS-CoV-2. Table 1 shows the results obtained by sample type from the study population.

### Sensitivity and specificity of plasma medium

Sensitivity was calculated as the number of true positive results by plasma as test sample compared with true positives by the reference swabs, and expressed as a percentage.

$$\text{True Positive} = 6; \text{ False Negative } = 69$$

$$\text{Therefore, sensitivity} = 6/6 + 69 \times 100 = 8\%$$

Specificity was calculated as the number of true negative specimens identified by using

**Table 1. Profile of SARS-CoV-2 RNA results based on sample type, analysed on BGI RT-qPCR molecular platform.**

| Sample type | OP/NP Swab | Plasma | Total | *P* value |
|---|---|---|---|---|
| | N (%) | N (%) | N (%) | |
| SARS-CoV-2 positive | 75 (60) | 6 (4.8) | 81 (32.4) | 0.003 |
| SARS-CoV-2 negative | 50 (40) | 119 (95.2) | 169 (67.6) | 0.068 |

**Table 2. Cycle threshold (Ct) values of COVID-19 swab samples in comparison with their corresponding plasma samples.**

| Ct values | Sample Type | |
|---|---|---|
| | OP/NP Swab (%) | Plasma (%) |
| 0 (Negative) | 50 (40) | 119 (95.2) |
| 12.0–18.9 | 19 (15) | 1 (0.8) |
| 19.0–25.9 | 27 (21.6) | 1 (0.8) |
| 26.0–32.9 | 20 (16) | 2 (1.6) |
| 33.0–38.0 | 9 (7.2) | 2 (1.6) |

plasma medium compared to true negatives by the swabs, and expressed as a percentage.

$$\text{True Negative} = 50; \; \text{False Positive} = 0$$

$$\text{Therefore, sensitivity} = 50/50 + 0 \times 100 = 100\%$$

The sensitivity and specificity of the RT-qPCR SARS-CoV-2 test with plasma was 8% and 100% respectively.

The Ct values of the six plasma positive SARS-CoV-2 samples were 35.9, 34.6, 16.4, 23.1, 32.6 and 32.2. (median Ct = 32.4). Ct values of swab samples were categorised and compared alongside the Ct values of the corresponding plasma samples as shown in Table 2.

## Clinical presentation

Out of the cases investigated, 55 (44%) were asymptomatic, while 70 (56%) had symptoms. The symptoms reported were categorised as mild (consisting of cough or fever or sore throat only), moderate (cough and/or fever and/or sore throat) and severe (shortness of breath, diarrhoea, chest pain with the other symptoms). Symptomatic mild, moderate and severe cases were 44 (35.2%), 18 (14.4%) and 8 (6.4%) respectively. Of the 75 persons positive by swab, only 32 (25.6%) had symptoms. The six persons positive by plasma all had mild symptoms. Those who were asymptomatic, negative by plasma but positive by swab were 25 (20%). Clinical presentations of the individuals that reported for COVID-19 testing are shown in Table 3.

**Table 3. Clinical presentation of individuals reporting for the SARS-CoV-2 assay.**

| Variable | Total (%) | SARS-CoV-2 swab Pos (%) | SARS-CoV-2 swab Neg (%) |
|---|---|---|---|
| Mean age (years): 41 | 125 (100) | 75 (60) | 50 (40) |
| Range (years): 18–72 | | | |
| **Sex** | | | |
| Male | 74 (59.2) | 43 (57.3) | 22 (44) |
| Female | 51 (40.8) | 32 (42.7) | 28 (56) |
| **Clinical Characteristics** | | | |
| Asymptomatic | 55 (44) | 25 (20) | 30 (24) |
| Symptomatic | 70 (56) | 32 (25.6) | 38 (30.4) |
| Mild: cough, fever, sore throat | 44 (35.2) | 19 (15.2) | 25 (19.9) |
| Moderate: fever, deep cough, fatigue, body aches | 18 (14.4) | 5 (4.0) | 13 (10.4) |
| Severe: Shortness of breath, diarrhoea, chest pain | 8 (6.4) | 8 (6.4) | 0 (0) |

## Discussion

PCR diagnostic testing, remains an important tool for informing patient management, saving lives and reducing SARS-CoV-2 spread with unavailability of proven effective therapy or vaccine, [15]. Community transmission is on the upward trend and majority of infections are asymptomatic hence, we could miss out such persons in blood donation centres, posing an infection risk. Information on COVID-19 transmission by blood is scarce.

There was a high (60%) prevalence of SARS-CoV-2 amongst the swab samples tested. The high prevalence may be due to the criteria utilised in selection of individuals for COVID-19 testing. Invitation for testing was at the time skewed towards those who filled on the online form that they were symptomatic and likely to have COVID-19. As time progressed, we realised that up to 80% of the population would by totally asymptomatic during the course of the infection. However, among their corresponding plasma samples, detection of SARS-CoV-2 RNA was low (4.8%). Other studies have corroborated the presence of viral RNA at very low levels in a fraction of blood samples from COVID-19 patients [21].

In this study plasma was shown to be an inefficient sample type to diagnose COVID-19. Exploring the use of antigen test may be a possibility to rule out infected samples that should be discarded from blood banks. In the course of the pandemic, when the virus becomes rooted (development of herd immunity), plasma may become significant as a source for antibody/antigen detection.

Over half (56%) of the study population were symptomatic. Most of the symptomatic cases observed were mild (35.2%) with a high rate of recovery. This is a fortunate occurrence in Nigeria and most of the African countries, as majority of people do not develop severe symptoms, and death rates are low, unlike in the temperate regions of Europe and North America [21].

In the current pandemic, community spread of COVID-19 is well documented, but transfusion transmission is not yet widely reported [16, 17]. This study highlights a potential implication of SARS-CoV-2 in donated blood in blood banking facilities. The outcome suggests need for further studies to verify the possibility of transmission via blood. In a study of Chinese blood donors, an insignificant number were reported SARS-CoV-2 RNA positive at minor levels [18]. It is likely that they may have been in pre-symptomatic stages of the disease. Virus concentrations during the viremic phase of the disease are low and transient. Monitoring safety of blood donations is of importance even though respiratory viruses are generally not transmitted by blood [13].

Recent studies have revealed SARS-CoV-2 viral shedding often occurs in early pre-symptomatic stages. This affects precision of diagnostic SARS-CoV-2 RT-PCR swab tests [19] especially if patients do not pursue molecular diagnostic testing until several days after onset of symptoms. The median plasma PCR CT value in this study was 32.4 (low CT) suggesting a low viral RNA concentration, in the range of between $10^2$ to $10^3$ copies per ml [20]. A recent study reported that the presence of viral RNA in blood samples is confined to a small percentage of acutely infected patients, and does not indicate the presence of infectious viral particles [21]. Another recent report involved a 21-year-old man with very severe aplastic anaemia, who received platelet transfusion from an individual who was later diagnosed with COVID-19. The recipient patient tested negative for COVID-19 showing evidence of no transmission [22]. Yet another report of COVID-19 patients presenting in the emergency department corroborates the findings in this present research. SARS-CoV-2 obtained by real-time PCR was found in six (5.9%) of 118 samples tested. This further gives credence to COVID-19 in blood but in few patients [23]. The American Association of Blood Banks (AABB) and Center for Disease Control and Prevention (CDC) currently do not recommend any specific SARS-CoV-2-related

actions by blood collection establishments [24]. The European centre for Disease Prevention and Control (ECDC) advises a precautionary delay from blood donation for 21 days, after possible exposure to confirmed patients. Likewise, those recovering from COVID-19 are to avoid donating blood for at least 28 days after symptom resolution [25]. It is better to err on the side of caution.

## Study limitations

The following limitations were observed in this study: 1. The findings from this study may not compare with blood bank individuals as these were suspected COVID-19 positive individuals. 2. The exact day of symptoms onset was not assessed. 3. We could not control for those that refused consent. 4. We could not decipher those who were not truthful upon filling the section on symptoms in the online forms.

## Conclusion

As recommended by WHO, molecular PCR testing for COVID-19 remains the gold standard while the nasopharyngeal and oropharyngeal swabs are the preferred samples for COVID-19 testing. Screening of plasma is not suitable to rule out SARS-CoV-2 infection. Further studies are required to determine the role of blood in COVID-19 transmission and the most suitable means of blood screening that may be required for blood banking facilities.

## Supporting information

**S1 Data. COVID-19 plasma data new.**
(XLSX)

## Acknowledgments

We thank the participants and clinical staff who are providing care during this pandemic.

## Author Contributions

**Conceptualization:** Azuka Patrick Okwuraiwe, Rosemary Ajuma Audu.

**Data curation:** Azuka Patrick Okwuraiwe, Joseph Ojonugwa Shaibu, Samuel Olufemi Amoo.

**Formal analysis:** Azuka Patrick Okwuraiwe, Ayorinde Babatunde James.

**Funding acquisition:** Babatunde Lawal Salako, Rosemary Ajuma Audu.

**Investigation:** Azuka Patrick Okwuraiwe, Joseph Ojonugwa Shaibu, Samuel Olufemi Amoo, Fehintola Anthonia Ige, Leona Chika Okoli, Abul-Rahman Ahmed, Jamda Ponmak, Judith O. Sokei.

**Methodology:** Azuka Patrick Okwuraiwe, Chika Kingsley Onwuamah, Joseph Ojonugwa Shaibu, Samuel Olufemi Amoo, Fehintola Anthonia Ige, Ayorinde Babatunde James, Leona Chika Okoli, Abul-Rahman Ahmed, Judith O. Sokei.

**Project administration:** Babatunde Lawal Salako, Rosemary Ajuma Audu.

**Resources:** Babatunde Lawal Salako, Rosemary Ajuma Audu.

**Supervision:** Chika Kingsley Onwuamah, Sulaimon Akanmu, Babatunde Lawal Salako, Rosemary Ajuma Audu.

**Validation:** Chika Kingsley Onwuamah, Sulaimon Akanmu.

**Visualization:** Azuka Patrick Okwuraiwe, Sulaimon Akanmu, Rosemary Ajuma Audu.

**Writing – original draft:** Azuka Patrick Okwuraiwe, Jamda Ponmak.

**Writing – review & editing:** Azuka Patrick Okwuraiwe, Joseph Ojonugwa Shaibu, Samuel Olufemi Amoo, Fehintola Anthonia Ige, Ayorinde Babatunde James, Leona Chika Okoli, Abul-Rahman Ahmed, Sulaimon Akanmu, Rosemary Ajuma Audu.

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
