## [Decision Letter · Decision Letter 0]

17 Feb 2021

PONE-D-20-37710

Low-level SARS-CoV-2 RNA detected in Plasma Samples from a cohort of Nigerians: implications for blood transfusion

PLOS ONE

Dear Dr. Azuka Patrick Okwuraiwe,

Thank you for submitting your manuscript to PLOS ONE. After careful consideration, we feel that it has merit but does not fully meet PLOS ONE’s publication criteria as it currently stands. Therefore, we invite you to submit a revised version of the manuscript that addresses the points raised during the review process.

This study is reasonable and the results mostly support the conclusion. However, further revision according to the comments from reviewers should be considered.

We look forward to receiving your revised manuscript.

Kind regards,

Wen-Wei Sung, M.D., Ph.D.

Academic Editor

PLOS ONE

Journal Requirements:

"This project was supported by the Nigerian Institute of Medical Research who purchased reagents for the study."

"The authors received no specific funding for this work."

4.Thank you for submitting the above manuscript to PLOS ONE. During our internal evaluation of the manuscript, we found significant text overlap between your submission and the following previously published works.

https://www.sciencedirect.com/science/article/pii/S1876034120304676?via%3Dihub

https://covidtestingproject.org/faq.html

https://ccpp19.org/healthcare_providers/virology/detecting_virus.html

https://journals.plos.org/plosone/article?id=10.1371%2Fjournal.pone.0233947

Please revise the manuscript to rephrase the duplicated text, cite your sources, and provide details as to how the current manuscript advances on previous work. Please note that further consideration is dependent on the submission of a manuscript that addresses these concerns about the overlap in text with published work.

Reviewers' comments:

Reviewer's Responses to Questions

**Comments to the Author**

1. Is the manuscript technically sound, and do the data support the conclusions?

Reviewer #1: Yes

Reviewer #2: Yes

Reviewer #3: Yes

2. Has the statistical analysis been performed appropriately and rigorously? 

Reviewer #1: Yes

Reviewer #2: Yes

Reviewer #3: N/A

3. Have the authors made all data underlying the findings in their manuscript fully available?

Reviewer #1: No

Reviewer #2: Yes

Reviewer #3: Yes

4. Is the manuscript presented in an intelligible fashion and written in standard English?

Reviewer #1: Yes

Reviewer #2: Yes

Reviewer #3: Yes

5. Review Comments to the Author

Reviewer #1: Well written paper, easy to understand and clear.

Conclusions are also clear.

Line 66: it says the correct area—define the correct area as readers vary in understanding.

Table 1: Reflect how you calculated the sensitivity and specificity. Could label a, b, c, d.

How did you try to control or could not control for the refusers so there are no biases? How will authors know if the refusers may have actually tested positive or negative. What is the size of those who did not consent? This data will be helpful. or at least include in study limitations.

Line 200: Not convinced that plasma will be helpful for detection though, can this be explained further ?

Reviewer #2: This is an interesting study comparing the ability to detect SARS-COV-2 RNA within the circulation to gold standard nasal/oral swabs. The authors demonstrate that nasal/oral swabs is more accurate than using blood plasma.

The abstract/results needs to be amended; "There was no false positive recorded, but 119 (95.2%) false negatives were obtained by plasma". I believe that this should be changed to; "There was no false positive recorded, but 69 (92.0%) false negatives were obtained by plasma" There reason for this is that the authors were only able to detect 75/125 using swabs (gold standard) and only therefore know that these were true positive COVID-19 samples. The remaining 50/125 should be classed as negative. This would also alter the sensitivity & specificity calculations. This should also be updated in the results section.

Since it is now Feb 2021, the prevalence figures in the introduction need to be updated (data is from July 2020).

It is a shame that the authors do not have any data on symptom onset or longitudinal sampling. This would have greatly enhanced the impact of the manuscript.

Reviewer #3: Review

Low-level SARS-CoV-2 RNA detected in Plasma Samples from a cohort of Nigerians: implications for blood transfusion

This study assesses the specificity and sensitivity of the PCR tests performed in plasma samples in comparison with oral and nasal swab samples to detect Sars-CoV-2 infection. They found that plasma PCR had 95.2% of false negatives. Although the possibility of covid transmission through blood has yet to be determined, these results might have implications for blood transfusion testing requirements.

1. In the Study population section, please, include a table with the demographic (age, sex, etc) and clinical characteristics of the patients (number and type of symptoms).

2. In page 6, line 120, it says that symptomatic and asymptomatic individuals were recruited; also, in page 9, line 181, it says that 20% of patients were asymptomatic, negative by plasma and positive by swab. However, in the Study population section (page 5, line 113) it says that the study criteria included presenting symptoms, such as fever, cough or breathing difficulty. Please, clarify whether there were asymptomatic patients and inclusion criteria.

3. What was the starting volume of plasma to extract RNA? Increasing the starting volume would probably increase the sensitivity of the test.

4. In the description of the PCR method, please, include device and PCR parameters.

5. What was the minimum CT to consider a sample positive for Sars-CoV-2?

6. If possible, it would be very interesting to look at the prevalence of antibodies against Sars-CoV-2 in the same cohort of patients.

7. In Table 1, please, add statistical analysis and p-values, and describe the statistics used in the Methods section.

8. In page 8, line 166, the Cts of the plasma positive samples are included; please, add also the Cts of the same samples that were obtained by nasal/oral swab to better compare both methods.

9. In Discussion, page 9, line 196, a preprint is cited. In general, citation of non-peer- reviewed articles should be avoided. If cited, clearly indicate in the main text that is a pre-print and has not been peer-reviewed.

10. At least another paper on the same subject has been published (J Clin Virol. 2020 Dec. Low prevalence of SARS-CoV-2 in plasma of COVID-19 patients presenting to the emergency department). Please, discuss.

11. Discussion of reports showing covid transmission from mother to newborn might be relevant for blood transmission.

Minor corrections:

1. Substitute the term “clients” by “patients”, “individuals” or “participants”.

2. Page 3, line 51: It says: “The virus was originated in bats”. This has yet to be proved. Better to say “likely”.

6. PLOS authors have the option to publish the peer review history of their article (what does this mean?). If published, this will include your full peer review and any attached files.

Reviewer #1: No

Reviewer #2: No

Reviewer #3: No

---

## [Author Response · Author response to Decision Letter 0]

26 Apr 2021

Response to Reviewers

PONE-D-20-37710

Low-level SARS-CoV-2 RNA detected in Plasma Samples from a cohort of Nigerians: implications for blood transfusion

PLOS ONE

Response: Done, manuscript presently meets PLOS ONE's style requirements, including those for file naming.

"This project was supported by the Nigerian Institute of Medical Research who purchased reagents for the study."

"The authors received no specific funding for this work." 

Response: Funding statement removed from manuscript. Update my Funding Statement to read, “The authors received no specific funding for this work”.

 Please include your amended statements within your cover letter; we will change the online submission form on your behalf. Response: Funding statement has been included in the cover letter.

3. Your ethics statement should only appear in the Methods section of your manuscript. If your ethics statement is written in any section besides the Methods, please delete it from any other section. Response: Ethics statement removed from other sections of the manuscript.

 4.Thank you for submitting the above manuscript to PLOS ONE. During our internal evaluation of the manuscript, we found significant text overlap between your submission and the following previously published works.

https://www.sciencedirect.com/science/article/pii/S1876034120304676?via%3Dihub

https://covidtestingproject.org/faq.html

https://ccpp19.org/healthcare_providers/virology/detecting_virus.html

https://journals.plos.org/plosone/article?id=10.1371%2Fjournal.pone.0233947

Please revise the manuscript to rephrase the duplicated text, cite your sources, and provide details as to how the current manuscript advances on previous work. Please note that further consideration is dependent on the submission of a manuscript that addresses these concerns about the overlap in text with published work.

We will carefully review your manuscript upon resubmission, so please ensure that your revision is thorough. Response: All duplicated texts have been rephrased; sources have been cited.

Reviewers' comments:

Reviewer's Responses to Questions

 Comments to the Author

1. Is the manuscript technically sound, and do the data support the conclusions?

Reviewer #1: Yes

Reviewer #2: Yes

Reviewer #3: Yes

 2. Has the statistical analysis been performed appropriately and rigorously?

 Reviewer #1: Yes

Reviewer #2: Yes

Reviewer #3: N/A

 3. Have the authors made all data underlying the findings in their manuscript fully available?

Reviewer #1: No

Reviewer #2: Yes

Reviewer #3: Yes

 4. Is the manuscript presented in an intelligible fashion and written in standard English?

Reviewer #1: Yes

Reviewer #2: Yes

Reviewer #3: Yes

 5. Review Comments to the Author

 Reviewer #1: Well written paper, easy to understand and clear.

Conclusions are also clear.

Line 66: it says the correct area—define the correct area as readers vary in understanding. Response: Done. Statement changed.

Table 1: Reflect how you calculated the sensitivity and specificity. Could label a, b, c, d. Response: calculations have been reconstructed.

How did you try to control or could not control for the refusers so there are no biases? How will authors know if the refusers may have actually tested positive or negative. What is the size of those who did not consent? This data will be helpful. or at least include in study limitations. Response: Done, see Study limitations.

Line 200: Not convinced that plasma will be helpful for detection though, can this be explained further? Response: Statement changed to reflect that plasma will only be useful in antibody and antigen detection assays.

Reviewer #2: This is an interesting study comparing the ability to detect SARS-COV-2 RNA within the circulation to gold standard nasal/oral swabs. The authors demonstrate that nasal/oral swabs are more accurate than using blood plasma.

The abstract/results needs to be amended; "There was no false positive recorded, but 119 (95.2%) false negatives were obtained by plasma". I believe that this should be changed to; "There was no false positive recorded, but 69 (55.2%) false negatives were obtained by plasma" The reason for this is that the authors were only able to detect 75/125 using swabs (gold standard) and only therefore know that these were true positive COVID-19 samples. The remaining 50/125 should be classed as negative. This would also alter the sensitivity & specificity calculations. This should also be updated in the results section. 

Response: Change effected, however, it did not alter the sensitivity and specificity.

Since it is now Feb 2021, the prevalence figures in the introduction need to be updated (data is from July 2020). Response: Figures updated

It is a shame that the authors do not have any data on symptom onset or longitudinal sampling. This would have greatly enhanced the impact of the manuscript. Response: Unfortunately, no data on those. but I could attempt that in a subsequent research.

Reviewer #3: Review

Low-level SARS-CoV-2 RNA detected in Plasma Samples from a cohort of Nigerians: implications for blood transfusion

This study assesses the specificity and sensitivity of the PCR tests performed in plasma samples in comparison with oral and nasal swab samples to detect Sars-CoV-2 infection. They found that plasma PCR had 95.2% of false negatives. Although the possibility of covid transmission through blood has yet to be determined, these results might have implications for blood transfusion testing requirements.

1. In the Study population section, please, include a table with the demographic (age, sex, etc) and clinical characteristics of the patients (number and type of symptoms). Response: Done, please see Table 3 in the text.

2. In page 6, line 120, it says that symptomatic and asymptomatic individuals were recruited; also, in page 9, line 181, it says that 20% of patients were asymptomatic, negative by plasma and positive by swab. However, in the Study population section (page 5, line 113) it says that the study criteria included presenting symptoms, such as fever, cough or breathing difficulty. Please, clarify whether there were asymptomatic patients and inclusion criteria. Response: The statement has been clarified.

3. What was the starting volume of plasma to extract RNA? Increasing the starting volume would probably increase the sensitivity of the test. Response: The starting volume was 200 µL. That is true.

4. In the description of the PCR method, please, include device and PCR parameters. Response: The device is included.

5. What was the minimum CT to consider a sample positive for Sars-CoV-2? Response: 13.0

6. If possible, it would be very interesting to look at the prevalence of antibodies against Sars-CoV-2 in the same cohort of patients. Response: Not possible to carry out at this time.

7. In Table 1, please, add statistical analysis and p-values, and describe the statistics used in the Methods section. Response: Done.

8. In page 8, line 166, the Cts of the plasma positive samples are included; please, add also the Cts of the same samples that were obtained by nasal/oral swab to better compare both methods. Response: Done in Table 2.

9. In Discussion, page 9, line 196, a preprint is cited. In general, citation of non-peer-reviewed articles should be avoided. If cited, clearly indicate in the main text that is a pre-print and has not been peer-reviewed. Response: Done, preprint removed.

10. At least another paper on the same subject has been published (J Clin Virol. 2020 Dec. Low prevalence of SARS-CoV-2 in plasma of COVID-19 patients presenting to the emergency department). Please, discuss. Response: Paper discussed and cited.

11. Discussion of reports showing covid transmission from mother to newborn might be relevant for blood transmission. Response: such discussion included.

Minor corrections:

1. Substitute the term “clients” by “patients”, “individuals” or “participants”.

2. Page 3, line 51: It says: “The virus was originated in bats”. This has yet to be proved. Better to say “likely”.

 6. PLOS authors have the option to publish the peer review history of their article (what does this mean?). If published, this will include your full peer review and any attached files.

Do you want your identity to be public for this peer review? For information about this choice, including consent withdrawal, please see our Privacy Policy.

Reviewer #1: No

Reviewer #2: No

Reviewer #3: No

---

## [Decision Letter · Decision Letter 1]

12 May 2021

PONE-D-20-37710R1

Low level SARS-CoV-2 RNA detected in plasma samples from a cohort of Nigerians: Implications for blood transfusion

PLOS ONE

Dear Dr. Azuka Patrick Okwuraiwe,

Thank you for submitting your manuscript to PLOS ONE. After careful consideration, we feel that it has merit but does not fully meet PLOS ONE’s publication criteria as it currently stands. Therefore, we invite you to submit a revised version of the manuscript that addresses the points raised during the review process.

We look forward to receiving your revised manuscript.

Kind regards,

Wen-Wei Sung, M.D., Ph.D.

Academic Editor

PLOS ONE

Journal Requirements:

Reviewers' comments:

Reviewer's Responses to Questions

**Comments to the Author**

1. If the authors have adequately addressed your comments raised in a previous round of review and you feel that this manuscript is now acceptable for publication, you may indicate that here to bypass the “Comments to the Author” section, enter your conflict of interest statement in the “Confidential to Editor” section, and submit your "Accept" recommendation.

Reviewer #2: All comments have been addressed

Reviewer #3: (No Response)

2. Is the manuscript technically sound, and do the data support the conclusions?

Reviewer #2: Yes

Reviewer #3: Yes

3. Has the statistical analysis been performed appropriately and rigorously? 

Reviewer #2: Yes

Reviewer #3: N/A

4. Have the authors made all data underlying the findings in their manuscript fully available?

Reviewer #2: Yes

Reviewer #3: Yes

5. Is the manuscript presented in an intelligible fashion and written in standard English?

Reviewer #2: Yes

Reviewer #3: Yes

6. Review Comments to the Author

Reviewer #2: (No Response)

Reviewer #3: Low level SARS-CoV-2 RNA detected in plasma samples from a cohort of Nigerians: Implications for blood transfusion

The authors have addressed most of my questions. Only two minor corrections need to be done:

1. Please, include the starting amount of plasma (200 ul) for RNA extraction is the Methods section.

2. Please, include the statistic test that was applied (Chi-square, Fisher exact test, etc) in Methods and in Results sections.

7. PLOS authors have the option to publish the peer review history of their article (what does this mean?). If published, this will include your full peer review and any attached files.

Reviewer #2: No

Reviewer #3: No

---

## [Author Response · Author response to Decision Letter 1]

14 May 2021

The authors have addressed most of my questions. Only two minor corrections need to be done:

1. Please, include the starting amount of plasma (200 ul) for RNA extraction is the Methods section.

Response: 200 ul has been added into the Methods section.

2. Please, include the statistic test that was applied (Chi-square, Fisher exact test, etc) in Methods and in Results sections.

Response: This particular study did not require rigorous statistics, except proportions.

---

## [Editor Report · Decision Letter 2]

19 May 2021

Low level SARS-CoV-2 RNA detected in plasma samples from a cohort of Nigerians: Implications for blood transfusion

PONE-D-20-37710R2

Dear Dr. Azuka Patrick Okwuraiwe,

We’re pleased to inform you that your manuscript has been judged scientifically suitable for publication and will be formally accepted for publication once it meets all outstanding technical requirements.

Kind regards,

Wen-Wei Sung, M.D., Ph.D.

Academic Editor

PLOS ONE

---

## [Editor Report · Acceptance letter]

2 Jun 2021

PONE-D-20-37710R2 

Low level SARS-CoV-2 RNA detected in plasma samples from a cohort of Nigerians: implications for blood transfusion 

Dear Dr. Okwuraiwe:

I'm pleased to inform you that your manuscript has been deemed suitable for publication in PLOS ONE. Congratulations! Your manuscript is now with our production department. 

Kind regards, 

on behalf of

Dr. Wen-Wei Sung 

Academic Editor

PLOS ONE